# Management of Solid Waste Containing Fluoride—A Review

**DOI:** 10.3390/ma15103461

**Published:** 2022-05-11

**Authors:** Małgorzata Olejarczyk, Iwona Rykowska, Włodzimierz Urbaniak

**Affiliations:** 1Faculty of Chemistry, Adam Mickiewicz University, ul. Uniwersytetu Poznańskiego 8, 61-614 Poznań, Poland; malgorzata.olejarczyk@amu.edu.pl (M.O.); obstiwo@amu.edu.pl (I.R.); 2Construction Company “Waciński” Witold Waciński, ul. Długa 15, 83-307 Kiełpino, Poland

**Keywords:** solidification/stabilisation, fluoride removal, defluorination techniques, adsorption, industrial waste

## Abstract

Technological and economic development have influenced the amount of post-production waste. Post-industrial waste, generated in the most considerable amount, includes, among others, waste related to the mining, metallurgical, and energy industries. Various non-hazardous or hazardous wastes can be used to produce new construction materials after the “solidification/stabilization” processes. They can be used as admixtures or raw materials. However, the production of construction materials from various non-hazardous or hazardous waste materials is still very limited. In our opinion, special attention should be paid to waste containing fluoride, and the reuse of solid waste containing fluoride is a high priority today. Fluoride is one of the few trace elements that has received much attention due to its harmful effects on the environment and human and animal health. In addition to natural sources, industry, which discharges wastewater containing F− ions into surface waters, also increases fluoride concentration in waters and pollutes the environment. Therefore, developing effective and robust technologies to remove fluoride excess from the aquatic environment is becoming extremely important. This review aims to cover a wide variety of procedures that have been used to remove fluoride from drinking water and industrial wastewater. In addition, the ability to absorb fluoride, among others, by industrial by-products, agricultural waste, and biomass materials were reviewed.

## 1. Introduction

According to the circular economy principles, issues related to the correct and effective management of production waste are currently among the fundamental problems [1,2,3].

The following article comprehensively presents various materials used to neutralize fluorine ions, including waste materials. Moreover, a new material proposed by the authors was presented here, which is made of two industrial waste materials that form a sorbent for fluorine adsorption, and after use, it can be used in building material.

Environmental pollution due to the mismanagement of solid waste is a global problem. Many publications on specific waste streams have been published in the scientific literature to quantify their environmental impact [4,5,6,7,8,9,10]. N. Ferronato [10], in a review work, assessed the global problems associated with various waste materials, pointing out how they affect the environment, what their relation is to human health, and how they influence sustainable development. The results shown by the authors provide a reference point for scientists and stakeholders to quantify comprehensive effects and plan integrated solid waste collection and treatment systems to make it easier to achieve sustainable development at the global level [10].

The efficient management and further utilization of waste materials becomes a significant problem for the industry and is growing as the amount of waste materials is increasing, and management costs are rising for both the industry and local administrations [11,12,13,14,15,16,17,18,19]. Therefore, recycling and reusing industrial waste and by-products are of great importance [20,21,22,23]. In fact, in reducing environmental problems and increasing economic benefits, there is a great need for technologies to transform waste materials into products of commercial value [24,25,26,27,28,29].

For example, some waste materials are converted in the scope of solidification and stabilization (S/S) processes. The solidification aids in changing the physical state of waste, from a liquid to a solid material, by encapsulation, thus decreasing the level of migration to the environment. The stabilization, by applications of some chemical reactions, migrates dangerous materials to less soluble or less toxic forms [30].

There are several S/S processes and methods proposed, tested, and implemented in practice [31,32,33,34]. These solutions, however, still need a lot of research to increase their effectiveness/performance and long-term effects [31]. The massive usage of S/S products, e.g., as construction materials, is still blocked by the potential risk of migrating the contaminants to the environment, including the toxic materials. New research, however, points out low leachability factors, which indicates that the S/S waste (contaminant source) can be regarded as an environmentally sustainable material with potential beneficial uses in construction [35].

Therefore, research and development is needed for the wide production and utilization of construction materials from various nonhazardous or hazardous waste materials [36].

In our opinion, special attention should be paid to waste containing fluoride, and the reuse of solid waste containing fluoride is a high priority today.

As one of the most extended elements on earth [37], fluorine (F) is widely used in the chemical industry, which in turn has produced large amounts of fluorine-containing hazardous waste. Fluoride, which is the most electro-negative element in the halogen family, is considered to be one of the main environmental pollutants due to its low biodegradability, high reactivity, and popularity [38]. One of the sources of introducing fluoride into the environment is the industry, which discharges sewage containing F− ions to surface waters and contributes to an increase in the concentration of fluoride in waters and environmental pollution [39].

Fluoride is one of the few trace elements that has received much attention due to its harmful effects on the environment and human and animal health [40,41,42,43]. Sabine [44] Guth et al. reviewed the available literature to critically assess the risks to human health from fluoride exposure, with a focus on developmental toxicity. Several factors, such as pH, alkalinity, chemical composition of aquifers, hardness, etc., determine the presence and concentration of fluoride in water resources [45,46,47,48,49,50,51]. 

Table 1 summarizes the review publications that have been published over the past decade on fluoride removal from both drinking and industrial wastewater, followed by the impact of fluoride waste to the environment and human health, and finally defluorination techniques. It presents the state of the art in the field of fluorinated waste management in one place. Moreover, it summarizes most of the techniques already proposed. The effectiveness of various materials for fluoride removal has been reviewed, taking into account key factors such as pH, initial fluorine concentration, surface area, particle size, and temperature, as well as the occurrence of counterions influencing the process of defluorination [39,52,53,54,55,56,57,58,59,60,61,62].

Natural and anthropogenic processes contribute to the release of fluorine compounds into the environment, causing the fluoride concentration in the soil to be much higher than the limit values, which is further followed by health and environmental problems in many regions of the world.

## 2. Anthropogenic Sources of Contamination with Fluorine Compounds

In many countries around the world, high levels of fluoride are the result of discharges of sewage polluted with fluoride [52].

Such wastewater is usually produced by industry: superphosphate fertilizers [63,64,65]; glass and ceramics production processes [66,67]; aluminium and zinc smelters [68,69,70]; steel production; uranium enrichment plants; coal-fired power plants; beryllium extraction plants; oil refineries [61,69,70,71,72]; the photovoltaic solar cell industry [61,73,74,75,76,77,78]; the production of high-tech silicon-based semiconductors [61,75,76,77,78]; and municipal waste incineration plants through HF emissions caused by the incineration of fluorinated plastics, fluorinated textiles, or CaF_2_ in sludge [79]. Fluorine is also used in electroplating. In addition, it is used as a melting point depressant in metallurgical furnaces in the smelting process. Water from mines can be a significant source of fluoride.

Chlorofluorocarbons (CFS) have been used extensively as gas in deodorants and coolants in refrigerators. However, due to their destructive effect on the ozone layer, some of these compounds are withdrawn from use. Fluoride also migrates to the environment due to the use of pesticides (e.g., cyhalothrin, fenfluthrin, and tefluthrin) [21]. It is also liberated into the environment in the brick production process [76].

It is estimated that about 30% of pharmaceuticals (including antibiotics, antidepressants, drugs against asthma, and atresia) are based on fluoride. The next big emitters of fluoride are cooling gases used in air conditioning, ventilation, and cooling devices contain fluorine in their composition [80,81]. Fluor is released into the atmosphere by burning hard coal, brown coal, and fuel oil. Then, industrial dust containing soluble fluorides and gaseous compounds (including HF) is emitted [82]. Wastewater from these industries has a higher F− concentration than natural waters, starting from ten thousand mg/L, and in the case of phosphate production, fluoride concentrations in wastewater can reach up to 3000 mg/L [83].

The combustion of biomass releases fluoride into the atmosphere, which is the main stream of this atmospheric pollutant, which has not been characterized before. The emission of fine particles (PM 2.5) of water-soluble fluorine (F−) from the biomass combustion was assessed at the Fourth Fire Laboratory in Missoula Experiment (FLAME-IV) using X-ray energy dispersive scanning electron microscopy (SEM-EDX) and ion chromatography with conductivity detection. Based on recent assessments of global biomass combustion, they estimated that biomass combustion releases 76 Gg F− per year into the atmosphere, with an upper and lower limit of 40–150 Gg F− per year. The estimated F− flux from biomass combustion is comparable to fluoride emission from coal combustion and other anthropogenic sources. These data show that biomass combustion is the primary source of fluoride released into the atmosphere in the form of fine particles that can be transported over long distances [37].

As the aforementioned fluoride-originated pollutants raise several health problems, the World Health Organization (WHO) determined the acceptable level of fluoride content in drinking water at the level of 1.5 mg/L [45]. However, the concentration of fluorides in industrial wastewater mostly exceeds these WHO guidelines, reaching even thousands of milligrams per litre [40,84,85]. Thus, fluoride pollution in the aquatic environment, caused by natural and artificial activities, has been a significant problem worldwide. Searching for new, effective ways to remove of fluoride-originated waste from water seems to be very important [60].

## 3. Selected Types of Reagents for Fluoride Removal

Several conventional techniques may be pointed here, such as adsorption [61,67,68,69,70,71,72,73,86,87,88,89,90,91], chemical precipitation [86,92], coagulation and precipitation methods [72,93,94,95,96,97,98,99], ion exchange [100,101,102,103,104,105,106,107,108,109,110,111], and electrocoagulation [69,77,86,112,113,114,115,116,117,118,119,120,121,122,123], as well as more advanced membrane processes [83,124,125,126,127,128,129,130,131], reverse osmosis [132,133,134], and electrochemical treatment [69,115,116,117,118,119,120,121,122,123]. In general, such compounds as CaCl2 and CaO are added to precipitate fluoride in wastewater.

Each method has its advantages and limitations and can be operated with the appropriate efficiency provided that the process parameters are properly selected to remove fluoride in the appropriate concentration range [26,31,122].

Large-scale industrial operations generate vast amounts of waste, the management of which can be a serious problem. An interesting possibility is to convert such waste into sorbents used for the water defluorination. Then, industrial waste becomes an adsorbent to remove fluoride from aqueous solutions [29]. Figure 1 shows selected types of industrial waste that are used as such adsorbents. 

Among the various methods of water defluorination (as mentioned earlier), adsorption is the most commonly used technique to remove fluoride. Fluorine is adsorbed on a barrier composed of a resin and some mineral particles. This method is efficient, simple, and cheap. These factors are especially important for developing countries [7,28,102]. The adsorbents may be also based on the biomass from plants, even being an agricultural waste, and several industrial by-products. These inexpensive materials help replace an expensive commercial adsorbent such as activated carbon, which again has a regeneration problem. Agricultural and industrial waste materials are available in massive amounts, Some of them are inexpensive and biodegradable, and thus environmentally friendly [123].

A wide range of adsorbents and their modifications were tested to remove fluoride from water [26]. These include activated carbon [83,124,125,126,127,128], activated alumina [129,130,131,132,135], bauxite [53,131,133,134,136,137,138,139,140,141,142,143,144,145,146,147,148,149,150,151], hematite [137,152,153,154,155,156], polymer resins [94,138,139,157], activated rice husk [125,140,141,158], brick powder [142], pumice stone [143,159,160,161], red earth, charcoal, brick, fly ash, serpentine [144,162,163,164,165], Moringa oleifera seed extracts [166], granular ceramics [167], chitin, chitosan and alginate [135,145,146,147,148,149,150,151,155,156,168,169,170,171,172,173,174], modified iron oxide/hydroxide [175,176,177,178,179,180,181], hydroxyapatite (HAP) [182,183,184,185,186], zirconium-modified materials and ceremonies [58,187,188,189,190,191,192,193,194,195,196,197], titanium adsorbent [198,199,200], schwertmannite [201], modified cellulose [202,203], clays [165,204,205,206,207], zeolite [57,208,209,210,211,212,213], and magnesium modified sorbent [106,118,202,214].

## 4. Industrial Waste, By-Product, and Biomass as Fluoride Adsorbents

Red mud is waste produced by the aluminium industry during alkaline processing, namely by the so-called Bayer process. The red sludge is strongly alkaline. The use of industrial wastes such as red sludge for defluorination will significantly reduce their volume for the problem of land removal, soil and groundwater contamination, and landscaping for alternative uses [53].

The removal of fluoride from water using red mud granular according to batch and column adsorption techniques is described by Tor et al. [215]. Cengeloglu et al. [165] have studied defluoridation by using red mud as such and acid-treated red mud by 5.5 M HCl for drinking purposes, and Wei et al. [216] have used modified red mud with AlCl_3_ (MRMA) and further modified by heat-activated red mud (MRMAH) as an adsorbent for the removal of fluoride from water. Lv et al. [217] have investigated zirconium hydroxide modified red mud porous material to remove fluoride from aqueous solutions. Soni et al. [218] have studied red mud for defluoridation of water collected from the Sitapura Industrial Area, Jaipur (Rajasthan). The results of a study to remove fluoride from red mud by electrokinetic treatment and the feasibility of this technique were presented by Zhu et al. [219].

All authors reported promising results in removing fluoride. Waste mud was recently found as one of the most promising adsorbents due to its extremely low cost and wide availability. This waste is an untapped resource and, in some cases, presents serious disposal problems, so using waste sludge to remove contaminants is an important application. The authors [220] tested three different forms of waste sludge for their fluoride removal efficiency: primary sludge, acid-treated sludge, and precipitated waste sludge [220]. The precipitated waste sludge showed a higher yield than the others [52].

Sujana et al. [221] investigated the defluorination limit of alum sludge, a waste product of the bauxite alum production process by adding sulfuric acid, which mainly contains aluminium oxide and titanium with a small number of undecomposed silicates. Nigussie et al. [222] investigated the removal of fluoride using the sludge formed during aluminium sulphate production (alum) from kaolin in the sulfuric acid process.

The potential of fluoride adsorption in drinking water treated with spent bleaching earth (SBE) was investigated by Mahramanlioglu et al. [223]. SBE is a solid waste generated during oil processing, as it contains mainly residual oil not removed by filter pressing. SBE applications were found very efficient [224] to adsorb fluorine from water in one of the Iran regions at concentrations ranging from 2.28 and 5.4 mg/L, pH 7, and processing time about 180 min [38].

Fly ash or coal ash, also known as UK Powdered Fuel Ash or Carbon Combustion Residue (CCR), is the product of coal combustion that consists of solid particles (fine particles of burnt fuel) that are driven from coal boilers along with exhaust fumes. The ash that falls to the bottom of the boiler combustion chamber (colloquially called the furnace) is called bottom ash. Singh et al. [53] have studied the defluoridation of groundwater of Agra city by means of fly ash (ATF).

The batch adsorption capacity of fly ash has been studied by Nemade [225]. He observed that fluoride adsorption decreased continuously between pH 2 to 12. Xue [226] observed that the high pH of the solution caused a slight turbidity of the filtered water, the effectiveness of defluorination increases with the increase of fluoride concentration in the inflow, and both the amount of sifted water and the effectiveness of the defluorination increase with an increasing temperature. Geethamani et al. [227] used calcium hydroxide-treated fly ash (CFA) to remove fluoride in a batch study. The removal of more than 80% was achieved with a 10 mg/L fluoride solution with an equilibrium contact time of 120 min and a dose of 3 g/L CFA. The maximum removal of fluoride was at pH 7 [53].

Ramesh et al. [228] investigated the ability to remove bottom ash fluoride in batch and column modes. Thus, 73.5% fluoride removal was achieved with a bottom ash dose of 70 mg/100 mL with an optimal contact time of 105 min. The maximum removal efficiency of 83.2% was observed at pH 6.

Zhang et al., in their work [229], characterized the mechanisms of the detoxification of water-soluble fluoride in bottom ash and the decomposition of fluorine during the combustion of spent potting material (SPL) in response to four calcium compounds CaSiO_3_, CaO, Ca(OH)_2_, and CaCO_3_, which converted NaF into low toxicity compounds, with a conversion range at the level of 54.24–99.45%.

The cenosphere is a light, inert, hollow sphere made mainly of silica and aluminium oxide, filled with air or an inert gas, usually produced as a by-product of coal combustion in thermal power stations. Xu et al. [230] investigated fluoride removal using magnesium-loaded fly ash cenospheres (MLC) prepared by the wet impregnation of fly ash cenospheres with a magnesium chloride solution.

The removal of fluoride with aluminium hydroxide-coated rice husk ash was investigated Ganvir et al. [183]. Rice husk ash is obtained by burning rice husk ash and unshelled husk, the latter two being relatively cheap and massively produced materials. Mondal et al. [231] investigated the capacity of activated rice husk ash (ARHA) by washing and drying rice husk ash from a rice mill at 100 °C for 8 h in an electric furnace and further crushing into 250 μm particles. The fluoride adsorption capacity of such obtained adsorbent was 15.08 mg/g in the batch and 9.5 mg/g in the column test.

Aluminium Treated Bagasse Fly ash (ABF) treated with aluminium for drinking water defluorination with an initial fluorine concentration of 1–10 mg/L, with a sorbent dose range of 1–20 g/L at pH 6.0 were tested by Gupta et al. [232].

Jadhav et al. [233] used maize husk fly ash as an adsorbent for eliminating fluoride from water, with the efficiency reaching 86% at a pH value of 2 and reaction time about two hours.

Waste carbon slurry for fluoride removal was investigated by Gupta et al. [234]. This compound is obtained from fuel oil-based generators of the fertilizer industry. The maximum fluoride adsorption capacity was reported at a level of 4.861 mg/g, with a reaction time of about one hour and pH equal to 7–8.

The ability of the adsorbent produced from coal-mining waste to remove fluoride from an aqueous solution was investigated by CInarli et al. [235]. The optimal pH for the reaction was found at the level of 3.5. To the same goal, Kumari et al. [236] used shale (coal mine waste) as a native shell (NS) adsorbent and heat-activated shale (HAS) at various temperatures ranging from 350 °C to 550 °C.

Islam et al. [237] investigated the basic oxygen furnace slag, produced by the steel industry, to remove fluoride from water. Basic converter slag (BOFS) mainly contains 46.5% CaO, 16.7% iron oxide, 13.8% SiO2, and some other components. The thermal activation of BOFS (TABOFS) by heating at 1000 °C for 24 h increased the porosity and surface area, leading to increased fluoride adsorption and resulting in fluoride removal at the level of 93% (in comparison with initial 70%).

Lai and Liu [238] used a spent catalyst (a by-product of the petrochemical industry) to remove fluoride from aquatic environments. This compound consists mainly of porous silica and alumina, and it is efficient enough to remove fluoride. Tsai and Lui [239] examined spent iron-coated catalyst by coating 0.1 and 0.5 M Fe(NO_3_)_3_ to remove fluoride from an aqueous solution. Fluoride adsorption decreased with an increasing pH. The fluoride adsorption reaction was endothermic, and the rate of reaction increased with temperature.

Bauxite is a basic source of such metals as aluminium and gallium. It is a sedimentary rock with a relatively high aluminium content. Das et al. [240] used a thermally activated titanium-rich bauxite (TRB) for the removal of the fluoride excess from drinking water. Lavecchia et al. [154] investigated bauxite with a high alumina content (81.5%) to remove fluoride from contaminated water. The percent removal of bauxite from fluoride in the pretest was 38.5%. Chaudhari [241] used bauxite to defluoridation water. It was observed that the optimal dose of the adsorbent was 1.8 g/50 mL, while the process took 90 min at the optimum pH of 6.0.

Bibi et al. used hydrated cement, brick dust, and marble flour to de-fluorine and remove arsenic from the water. The presence of co-anions did not significantly affect the effectiveness of arsenic and fluoride removal [53]. Kang et al. [242] investigated Cement Paste for removing fluoride as a low-cost solution. The cement paste was competitive with lime, the prevalent fluoride-removing agent [52].

Zhang et al. [186] investigated the possibility of removing fluoride using recycled phosphogypsum. The latter was applied in the form of HAP nanoparticles using microwave radiation technology [52].

Oguz used lightweight concrete (building material) [243] as an adsorbent to remove fluoride from water, and its effectiveness was tested. The maximum adsorption of fluoride took place at pH 6.9. Additionally, hydrated cement [244] and hardened alumina cement [245] were tested to remove fluoride from an aqueous solution. Various forms of apatite have been used to remove fluoride because it has shown good defluorination prospects, namely synthetic nano-hydroxyapatite (n-Hap), biogenic apatite, processed biogenic apatite, and geogenic apatite [246]. The fluoride adsorption was determined to decrease with increasing concentration levels and pH value. Ultrasonic and microwave treatment also increased the effectiveness of the fluoride removal process [247,248]. The influence of low molecular weight organic acids (LMWOA) on the defluoridation capacity of nano-hydroxyapatite (nHAP) from an aqueous solution was investigated [249]. Cellulose nanocomposites @ hydroxyapatite (HA) were prepared in NaOH/thiourea/urea/H_2_O by in situ hybridization [203]. Aluminium-modified hydroxyapatite (Al-HAP) was also used for defluoridation [250]. High-purity phosphogypsum (PG) nanoparticles were also used, showing an excellent fluoride adsorption capacity [54,186]. 

Waste clay brick (WCB) is a silicate solid waste, the recycling of which is of significant environmental and social importance. WCB is used in the production of concrete and mortar, as a raw material, or an additive for the production of secondary cement.

In recent years, more and more attention has been paid to the recycling of waste clay bricks, and the extension of their recyclable use has laid a solid foundation for improving its utility value [55,251].

Bleaching powder, also known as chlorinated lime (calcium oxychloride), is mainly composed of calcium hypochlorite. It is widely used as a disinfectant for drinking or swimming pool water and as a bleaching agent. The whitening powder generally has advantageous properties as an economical and viable replacement for other adsorbents for removing fluoride from an aqueous solution. In addition to being a disinfectant, it also acts as a defluorant. Kagne et al. [252] used a bleaching powder to remove fluoride, raising the removal ratio from to 90.6% [52].

Li Wang et al. [84] adopted the new calcium-containing calcite precipitating and assisted precipitating fluorspar to treat wastewater containing fluoride. Key parameters of the reaction were determined, such as reaction timing, the rate of the oscillation, the doses of hydrochloric acid and calcite, etc. 

Chen et al. [253] developed a ceramic-based adsorbent for removing fluoride from an aqueous solution. The adsorbent showed sufficient mechanical resistance for long-term adsorption, as well as high efficiency. The same authors also reported results of batch tests of fluoride removal using a surface-modified granular ceramic with an Al-Fe complex [52].

Detailed information on the above-mentioned adsorbents is presented in Table 2.

## 5. Fluoride Wastes Removal in Industrial Processes

### 5.1. Industrial Production of Aluminium Fluoride

The mass production of aluminium fluoride forces significant amounts of silica gel being waste-contaminated with fluoride ions [24,255]. For example, a main fertilizer producer in Lithuania, a joint-stock company “Lifosa”, generates approximately 15 thousand tons per year of the mentioned waste during the manufacture of 17 thousand tons of AlF3 [256]. AlF3 is formed in the reaction of neutralizing hexafluorosilicic acid with aluminium hydroxide. However, due to the strong bonding of fluoride ions to the crystal structure of the latter compound, the purification of silica gel waste is a challenge. As a result, the waste silica gel is mainly disposed in the landfill with no further treatment [25,26,255,257]. The long-term storage of this king of waste provokes many environmental problems, due to the fact of the leaching of fluoride into water [24,255]. According to the literature [258,259,260,261,262,263], the amount of toxic compounds may be reduced by removing them from waste or by reducing their mobility to the environment [11].

### 5.2. Industrial Waste from Semiconductor Factories

In recent years, the industrial production of electronic materials has contributed to an increase in the global concentration of fluoride and water pollution. The significant contributors to fluoride-contaminated wastewater are semiconductor manufacturers and industrial plants producing hydrofluoric acid, photovoltaic materials, plastics, and textiles [69].

It is assumed that almost 30% of waste produced in the semiconductor industry is of fluorine origin; thus, new treatment methods for this kind of waste are welcome, with one of them described in [65]. The authors converted fluoride waste to AlF3 by an aluminium treatment. AlF3 is then dissolved, and, at the same time, a calcium conditioner is added to replace the AlF3 with CaF2. The method is able to reduce the amount of fluorine contents by a factor of 75%. Moreover, the aluminium component is reusable, therefore the cost of the method is reasonably small [264].

Chemical vapor deposition (CVD) processes are widely used in the production of solar cells and include the deposition of crystalline silicon from chlorosilanes, iodides, bromides, and fluorides [265]. An undesirable side effect is the release of toxic SiO_2_. The by-products of silicon film deposition consist of large amounts of SiO_2_ powder, HF vapours, SiH_4_, and PH_3_ [266]. These by-products are usually transported to the factory’s central scrubber or dust filter, and treatment produces large amounts of hazardous fluoride-containing sludges.

The effective and cheap treatment of fluorine-containing sludge resulting from CVD processes collected after cleaning the filter cartridge in a photovoltaic installation is located in southern Italy, as found by Zueva et al. [267]. 

In addition, the treatment of waste with alumina, magnesium sulphate, and lime was tested. These studies aimed to remove the F- content from the liquid phase of the sludge and examine the possibility of producing non-hazardous solid waste. Therefore, the toxicity characterization leaching (TCLP) procedure of the obtained solids was performed with and without thermal treatment. The best conditions for removing fluoride from liquid waste and converting the sludge into non-hazardous waste were related to water treatment with lime and magnesium sulphate.

Electrocoagulation coupled with flotation to treat semiconductor production wastewater was proposed by Hu et al. [77]. The fluoride ions were partially removed by precipitation with calcium in an electrolyser, to which sodium dodecyl sulphate was added to increase flotation. These treatments were effective in reducing fluoride and suspended solids in the wastewater. They lowered the concentration of fluoride from 806 mg/dm^3^ to 5–6 mg/dm^3^.

An original fluidization process to recover CaF_2_ from a synthetic fluoride solution was developed by Aldaco et al. [268]. Granulated calcite and silica were used as seed materials to recover the calcium fluoride by crystallization in a fluidized bed reactor. The inlet concentration of fluoride was 250 mg/L, and the final fluoride conversion was 92%, with a CaF_2_ content in the solid greater than 97% by weight. This process offers a good alternative for reducing solid waste and reusing calcium fluoride. 

In the work of Shin et al. [269], more than 99 wt.% precipitated HF and silicon during the pre-treatment of the solution and recovered Na_2_SiF_6_ to commercial grade 98.2%. The remaining solution contained 279 g/L acetic acid, 513 g/L nitric acid, and some HF. It was extracted with 2 ethylhexyl alcohol. Acetic acid was removed from the organic phase with deionized water to give 96.3% acetic acid recovery.

The recycling of SiO_2_-CaF_2_ nanoparticle sludge recovered from the semiconductor industry wastewater treatment was investigated by Lee and Liu [270]. The dried and powdered sludge was replaced with 5 to 20 wt.% Portland cement in mortar. The compressive strength of the modified mortar was higher in comparison with the fresh cement mortar after three days of hardening. Moreover, the Toxicity Trait Leaching Procedure (TCLP) showed that no heavy metals were released from the modified mortars. In another study, similar results were obtained with a different deposit produced in the polishing operations of the IC industry. This sludge, consisting of hazardous compounds such as SiO_2_, Al_2_O_3_, CaF_2_, and unknown organic compounds, was used to replace 10 wt.% cement powder to produce concrete. The compressive strength was comparable to that of regular Portland cement, while the TCLP test did not detect any metal release [271]. In another study, Lee [270] investigated the addition of a PV sludge/fly ash slag mixture for the production of cement mortar. The optimal mixture, determined by the Taguchi method, was 20.9 wt.%. cement flour, 4.3% volatile slag, 3.4% PV sludge, and 71.4% sand. The optimally modified cement mortar showed an increased compressive strength from the fourth day of maturation, reaching the maximum value of 132% after 7 days in relation to the compressive strength of the mortar composed of fresh Portland cement.

As such, recycling sludge from the PV industry is essential to preventing their potential environmental hazards and helping to reduce the cement industry’s carbon footprint and environmental impact. 

One type of hazardous waste was the fluorine-containing sludge from the semiconductor industry, the safe treatment and disposal of which were ineffective. Da et al. [272] presented the research results on the assessment of the possibility of adding fluorine-containing sludge to cement clinker. The authors inform that the addition of 2.0% of the sludge significantly improved the flammability of the clinker and improved a formation of alite. However, increasing the amount of the sludge to 5.0% caused the profuse formation of interstitial phases and slowed down the formation of alite and belite. The presence of fluorite was high in the silicate phase, resulting in the accumulation of this compound mainly at the surface. The fluoride was immobilized by calcium, with the immobilization rates for fluorine, copper, zinc, and nickel reaching a level of 99.5%. A sludge addition did not cause any threats or side effects [272].

## 6. A New Concept(s) for the Production and Management of Fluoride Adsorbents

According to the review, there are many methods of removing fluoride, including using sorbents made from industrial waste, by-product, and biomass. Many sorbents and their application methods have been developed, dedicated to specific industrial processes, in which there are fluorides in the form of wastewater or waste. There are also known methods of using fluoride-containing wastes to produce new products used in many fields, including construction. However, further research on developing new solutions is still being carried out. One of the latest proposals is the production of composites from several types of waste, which, although they can be used alone as materials for removing fluorides, especially from water and sewage, must first be deeply processed; e.g., calcination or their direct use is associated with technical problems at the stage of separating the used sorbent from the treated liquid by means of filtration or sedimentation [273].

Paper sludge (PS) is generated as industrial waste in the process of recycling paper products, with the amount continuously increasing year by year [274]. PS mainly contains cellulose fibres (up to 50–60%) and inorganic fillers along with coating materials such as calcite, kaolinite, and talc [275]. The paper industry is of great importance to the natural environment due to the amount of PS produced and its disposal. A small part of PS waste is used in agriculture as a soil improver and fertilizer [276,277,278,279,280]. However, PS is mostly disposed in open landfills with no further treatment, which is a growing problem especially for highly developed countries. Recently, we observed some tries to use PS as an additive to cement [281], metakaolin for the production of ceramics and glass [282], fuel for energy recovery [283,284], and carbon adsorbent for removing organic pollutants [285], thus reducing the amount of PS disposed in landfills.

Takaaki Wajima et al. converted PS into an effective fluoride sequestrant by the process of calcination in a high temperature for several hours. They determined that PS fired at 800 °C shows highest fluorine absorption. The authors also pointed out the fact of the selective removal of fluoride in several solutions containing chlorides, nitrates, and sulphates [274].

Using paper slurries obtained in the creation of composites based on non-calcined sludge and post-soda lime is an interesting and innovative solution to remove fluoride from water and sewage [273]. Post-soda lime is a by-product formed in the process of separating the solid phase present in the still liquid during the production of soda by the Solvay method. The mixture mainly contains some calcium compounds (CaCl_2_, CaCO3, CaSO_4_, Ca(OH)_2_), magnesium and silica, sulphur, and aluminium. It is characterized by a very high hydration (up to 60%) and low particle size distribution (less than 2 μm). Unfortunately, such properties strongly limit the traditional usage of this waste material [286]. According to the invention, the above problems are solved by the method [287,288], where very fine soda ash, preferably from clarifiers, is applied to fibrous cellulosic support in the form of papermaking sludges. The result is a composite that is highly permeable to liquids and removes fluoride ions very efficiently, with the same amount of added fluoride precipitant being even twice as high as in the case of traditional materials used to remove fluorides (such as ground limestone and chalk). The content of cellulose fibres in the composite allows it to be shaped practically in any form, e.g., in the form of granules, pellets, flat membranes, plates, cylinders, etc., depending on the user’s needs. They can also be placed in filter bags that are permeable to solutions, so they can be used repeatedly until the calcium compounds are entirely converted. After use, they can be easily removed from the treated solution. The used sorbent can be used analogously to paper sludge, for example as an additive/filler for building materials [36]. It was shown that, due to the calcium compounds used, the targeted material may be supplemented with hazardous mineral waste containing fluorine, e.g., in form of post-crystallization lye formed in the processing of fluosilicic acid or phosphogypsum. The approach improves the reuse of waste materials, as well as minimizes the usage of raw materials, contributing to the concept of a circular economy.

## 7. Conclusions

The negative impact of hazardous waste on the health of ecosystems, including humans, is becoming a rapidly growing global problem. The large amount of waste materials caused by the industry and human life is becoming a huge problem for both enterprises and local administrations, increasing the costs of everyday activities. Therefore, recycling and reusing industrial waste and by-products are of great importance. Many of them can be used to produce new construction materials after “solidification/stabilization” processes. Such materials may be used as admixtures or raw materials. In this case, an assessment of the leaching of the contaminants should be of particular concern, as well as the overall efficiency of the conversion process. Therefore, it is important to know the properties and conditions of use of fluoride binders.

Several materials have been proposed and tested as adsorbents towards efficient fluoride removal, taking into account also low processing costs and minimal side effects. The research in this area is undergoing, and the authors describe a high adsorption of fluoride-originated waste. However, the proposed methods usually depend on the particular pH and other process parameters that are difficult to achieve and maintain. Moreover, the adsorbents usually cannot be fully reused without costly regeneration. In addition, the competing ions show an affinity to the same active parts of the adsorbent, and the excess of some organic compounds delays the process balance.

In general, the concentration of fluoride ions can be reduced by a number of methods. Research on new methods for removing fluoride ions is still ongoing. At the same time, efforts are made to increase the efficiency of existing technologies. The removal of fluoride ions is a significant problem because they have a negative impact on human health [8]. Extensive research is required to develop and implement low-cost, sustainable hybrid technologies that can overcome the disadvantages of stand-alone processes.

## Figures and Tables

**Figure 1 materials-15-03461-f001:**
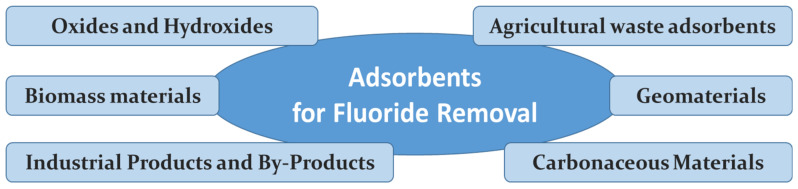
Selected types of industrial waste that are used as fluoride adsorbents.

**Table 1 materials-15-03461-t001:** A summary of review publications that have been published over the past decade on the removal of fluoride from drinking water and industrial wastewater.

Authors	Title	Aim
Habuda-Stanić M. et al., 2014 [52]	Review on Adsorption of Fluoride from Aqueous Solution	A list of various adsorbents (oxides and hydroxides, biosorbents, geomaterials, carbonaceous materials, and industrial by-products) and their modifications is discussed. This survey showed that various adsorbents, especially binary and trimetal oxides and hydroxides, have good potential for fluoride removal from aquatic environments.
Waghmare S.S. et al., 2015 [53]	Fluoride removal by industrial, agricultural and biomass wastes as adsorbents: a review	Reviews the fluoride uptake capacities of industrial by-products, agricultural wastes, and biomass materials from plants, grass, etc., and their modified forms as adsorbents in batch and column performance.
Tomar V. et al., 2013 [54]	A critical study on efficiency of different materials for fluoride removal from aqueous media	An extensive list of adsorbents for fluoride removal is compiled. In particular, nanomaterial-based adsorbents might be promising adsorbents for environmental and purification purposes.
Kumar P.S., 2019 [39]	Treatment of fluoride-contaminated water: a review	Reviews the origin of fluoride, the analysis of fluoride derivatives, and the technologies to remove fluoride from water, using different adsorbent types.
Nagendra Rao C.R. 2003 [58]	Fluoride and environment—a review	Current information on fluoride presence in the environment and its effects on human health, as well as basic methods of defluoridation.
Schlesinger W.H. et al., 2020 [59]	Global Biogeochemical Cycle of Fluorine	Synthesis of what is currently known about the natural and anthropogenic fluxes of fluorine.
He J. et al., 2020 [60]	Review of fluoride removal from water environment by adsorption	The recent developments in fluoride removal from the water environment by adsorption methods. Based on the review, four technical strategies of adsorption method, including nano-surface effect, structural memory effect, anti-competitive adsorption, and ionic sieve effect, were proposed.
Bhatnagar A. et al., 2011 [61]	Fluoride removal from water by adsorption—a review	An extensive list of various adsorbents from literature has been compiled, and their adsorption capacities under various conditions (pH, initial fluoride concentration, temperature, contact time, adsorbent surface charge, etc.) for fluoride removal are presented.
Bodzek M. et al., 2018 [39]	Fluorine in the Water Environment-Hazards and Removal Methods, Engineering and Protection of Environment	Detailed information on recent researchers’ efforts in the field of fluoride removal during potable water production. The contaminant elimination methods have been broadly divided in three sections, i.e., coagulation/precipitation, adsorption, and membrane techniques. Both precipitation with the use of calcium salts or coagulation with aluminium sulphate and ferric salts followed by sedimentation are used for fluorine removal. In electrocoagulation, a coagulant is generated in situ by means of oxidation of anode usually made of aluminium or iron.
Wang L. et al. 2019 [62]	A Review on Comprehensive Utilization of Red Mud and Prospect Analysis	Comprehensive utilization methods for reducing red mud (RM) environmental pollution and divides the comprehensive utilization of RM into three aspects: the effective extraction of valuable components, resource transformation, and environmental application.

**Table 2 materials-15-03461-t002:** Detailed information on the adsorbents used for fluoride removal.

Adsorbent	Concentration Range (mg/L)	pH Range	Contact Time (min)	Model Used to Calculate Adsorption Capacity	Maximum Adsorption Capacity (mg/g)	Ref.
Waste mud	-	2–8	0–480	Langmuir and Freundlich	27.2	[220]
Red Mud	5–150	4.7	15–540	Freundlich	0.851	[215]
	5	4.7	360	Redlich–Peterson and Freundlich	0.644	[215]
	100–1000	5.5	120	Langmuir and Freundlich	3.12 and 6.29	[165]
Modified red mud with AlCl_3_ (MRMA), heat activated red mud (MRMAH)	-	7–8		Langmuir	MRMA-68.07 MRMAH-91.28	[216]
Zirconium hydroxide modified red mud porous material Zr-modified RMPM	-	3	60	pseudo-second-order rate kinetics and pore diffusion models	0.6	[217]
Red mud	-	5.5	120	-		[218]
Alum sludge	-	5.5–6.5	-	-	5.35	[221]
Sludge produced during the manufacturing of aluminium sulphate (alum) from kaolin	10	3–8	-	-	332.5	[222]
Spent Bleach Earth (SBE)	-	3.5	-	-	7.75	[223]
Fly ash A and S	-	-	-	Freundlich	1.22 (A) 1.01 (S)	[226]
Calcium hydroxide treated fly ash (CFA)	10	7	120		10.86	[227]
Bottom ash	-	6	105	BDST	16.26	[228]
Magnesia-loaded fly ash cenospheres (MLC)	10	-	-	Thomas	5.884	[230]
aluminium-treated bagasse fly ash (ABF)	1–10	6	300	-	10	[232]
Maize husk fly ash	2.0 g/50 mL	2	120	Redlich-Peterson		[233]
Activated tea ash (AcTAP)		6	180	Langmuir	8.55	[231]
Waste carbon slurry obtained from fuel oil	15	7.58	120	Langmuir	4.861	[234]
Coal mining waste	-	3.5	-	Langmuir	15.67	[235]
Shale (coal mine waste) in the form of native shale (NS) and heat activated shale (HAS) at 350 °C, 450 °C and 550 °C	10-HAS550	3	24 h	Langmuir	0.358	[236]
Blast furnace slag generated from steel industry	10 mg/l	6–10	35	Langmuir	8.07	[237]
Spent catalyst (a by-product of petrochemical industry)	-	4	-		28	[238]
Iron coated spent catalyst	-	5.5–6.0	-	Langmuir	7.2–20.7	[239]
Thermally activated titanium rich bauxite (TRB)	10	5.5–6.5	-	Langmuir	3.8	[240]
High alumina (81.5%) content bauxite	-	-	-	Freundlich	3.125	[243]
Bauxite	10	6	90	Freundlicha, Langmuira Tempkina,	3	[241]
Hydrated cement (HC), brick powder (BP) marble powder (MP).	30	7 8 7	60	Langmuir	1.72 0.84 0.18	[254]
Bleaching powder	-	6–10	-	-	-	[244]
Rice husk ash, which was coated with aluminium hydroxide	10–60	7	60		15.08	[183]
Activated rice husk ash (ARHA)			100	Langmuir	0.402	[231]
Ceramic adsorbents consisting of Kanuma mud, with zeolite, starch, and FeSO_4_·7H_2_O	20–100	4–11	0–48 h	pseudo-second-order	2.16	[253]
Porous granular ceramic adsorbents containing dispersed aluminium and iron oxides	10	4–9	48 h	Langmuir and Freundlich	1.79	[249]
Iron-impregnated granular ceramics		7, 4		Langmuir and Freundlich	-	[167]
Recycled phosphogypsum in a form of HAP nanoparticles		7		Langmuir-Freundlich	19.742–25 °C 26.108–35 °C 36.914–45 °C 40.818–55 °C	[186]
HAP-calcium phosphate based bioceramic	-	-	-	Langmuir and pseudo-second-order	32.57	[250]
HAP Apatitic tricalcium phosphate.	up to 20 up to 60	4.16 4		Langmuir Langmuir	13.88–25 °C 14.70–30 °C 15.15–37 °C	[118,119]

## Data Availability

Not applicable.

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
