# Peer review of "Management of Solid Waste Containing Fluoride—A Review"

_materials, 2022, doi:10.3390/ma15103461_

Round 1

Reviewer 1 Report

It is an actual topic for research world, and a relevant topic for the journal.  A huge number of references is included in a reference list. In general, review articles is a good summary source of state-of-the-art for future research. Nevertheless, for this particular article to become such a good summary, a number of problems with the text of the manuscript, and/ or unclear issues still need to be solved:

  • The aim and scope of the review article should be clearly summarized in the Introduction.
  • It remained unclear how have you chosen references to be reviewed? Is it everything based on sources that you have found in review articles presented in Table 1? If so: is your article a review of reviews (or, summary of summaries)? Is there something that you have included extra to what is summarized in the previous review articles? Please clearly communicate the added value of your own review.
  • The writing style requires improvements. Instead on numerous statements on who did what, I propose to write from the point of view on what is known/ has been achieved/ what is still unclear on different aspects that you touch them in the manuscript. 
  • Use more Tables. For example, Table 2 is OK, but the text before it - not really. Please summarize tendencies of the adsorbents being used and/ or tested, comment the tendencies on their suitability/ efficiency, provide summarizing insights.
  • Be careful with providing links to reference sources: pay attention to usage of brackets, commas, lines, spaces, location of link in the sentence, etc.
  • Some further comments are provided directly in the text.

Author Response

Comments and Suggestions for Authors
It is an actual topic for research world, and a relevant topic for the journal. A huge number of
references is included in a reference list. In general, review articles is a good summary source of
state-of-the-art for future research. Nevertheless, for this particular article to become such a good
summary, a number of problems with the text of the manuscript, and/ or unclear issues still need to be
solved:

Thank you for all your comments.

ï‚· The aim and scope of the review article should be clearly summarized in the Introduction.

We agree with your suggestion. At the beginning, we have clarified what the article is about.

ï‚· It remained unclear how have you chosen references to be reviewed? Is it everything based on
sources that you have found in review articles presented in Table 1? If so: is your article a review of
reviews (or, summary of summaries)? Is there something that you have included extra to what is
summarized in the previous review articles? Please clearly communicate the added value of your own
review.

We also tried to address the newest research (including our own proposals), not cited in the above-
mentioned reviews, to provide up-to-date situation in the field of fluoride-waste management and
applications, such as 33-36, 94, 276, 282, 294-295. So, our paper is certainly not a review of reviews.

ï‚· The writing style requires improvements. Instead on numerous statements on who did what, I propose
to write from the point of view on what is known/ has been achieved/ what is still unclear on different
aspects that you touch them in the manuscript.

According to your suggestion, we have corrected the style, format and contents of the text, adding a
new section Section 5 to clarify our point of view.

ï‚· Use more Tables. For example, Table 2 is OK, but the text before it - not really. Please summarize
tendencies of the adsorbents being used and/ or tested, comment the tendencies on their suitability/
efficiency, provide summarizing insights

We agree with your suggestions. We improved the style of writing, but we decided that the next table
is not necessary and we included the conclusions resulting from the state of the art in Section 5.

ï‚· Be careful with providing links to reference sources: pay attention to usage of brackets, commas, lines,
spaces, location of link in the sentence, etc.

Thank you for this comment. Citations have been improved as suggested.

ï‚· Some further comments are provided directly in the text.

Replies to comments and suggestions marked in the text have been entered or changed directly in the
paper.

Reviewer 2 Report

Please revise it according to the following comments:

  1. This journal is committed to engaging with a wider public in order to promote the potential benefits of cutting-edge engineering research. Please describe in specific terms the potential impact of your work on the wider public. 
  2. Ideally, this article demonstrates how research results can be used in process engineering design and practice. Please outline briefly the engineering aspects in your paper (as opposed to scientific aspects).
  3. In what way does the work contribute to the SDGs? What are the trends and challenges of the technological approaches of the work based on the SDG paradigm?

You should think about how transformational the work is likely to be should be made so that the outcome of the work will have an impact on the community/society facing given sustainability-related challenges?

4. Write the practical applications of your work in a separate section, before the conclusions, and provide your good perspectives.

5. What are the bottlenecks of this work and how did you mitigate the impacts attributed to them?

  1. What did the technological gaps of the review focus on?
  2. What is the novelty (originality) of the review? And What is new in your review that makes a difference in the body of knowledge? 
  3. How did you do quality control (QC) and quality assurance (QA) on the obtained secondary data to validate the conclusions?
  4. What are possible technology-oriented applications of the work for commercialization purposes?
  5. How would this work advance the previous work done in the existing field of study and/or across other fields?
  6. Based on the data obtained, what are the implications of this work (a) to the field of study, (b) to industry, (c) economy, and/or (c) to the wider public/society in general?
  7. How would the outcomes of work directly contribute to global climate change mitigation and circular economy?
  8. What are the likely research impacts of this work globally, nationally and locally?
  9. What are the economical benefits of this work for industrial purposes?
  10. Make a table of comparison of this present work and other similar techniques in terms of operational parameters and operational cost; afterward, please give critical analysis on its technical feasibility and applicability for upscaling this treatment process.

16. Why do you believe your work to be important? What long-term impacts will it have on environmental protection and the wider public or the field following the completion of the project?

17. What about the thermal stability of the material obtained from the published studies?

18. A comprehensive managerial insight should be provided in this paper.

Author Response

Thanks for all comments. Your comments have been included in the text. Please see the attachment

  1. Ideally, this article demonstrates how research results can be used in process engineering design and practice. Please outline briefly the engineering aspects in your paper (as opposed to scientific aspects).

We have clarified our point of view in the new section No 5.

       2. In what way does the work contribute to the SDGs? What are the trends and challenges of the technological approaches of the work based on the SDG paradigm?

We have clarified this in the end of the new section No 5 (the description of the patents).

  3. You should think about how transformational the work is likely to be should be made so that the outcome of the work will have an impact on the community/society facing given sustainability-related challenges? This journal is committed to engaging with a wider public in order to promote the potential benefits of cutting-edge engineering research. Please describe in specific terms the potential impact of your work on the wider public. 

We have clarified this in the end of the new section No 5.

  1. Write the practical applications of your work in a separate section, before the conclusions, and provide your good perspectives.

We have posted the indicated comments in the section No 5.

  1. What are the bottlenecks of this work and how did you mitigate the impacts attributed to them?

We introduce several changes that were marked in the text.

6. What did the technological gaps of the review focus on?

Our review is a complete up-to-date state of technology. It was also an initial state for us to propose some new solutions and to compare them with the previously described. Due to some restrictions of the patenting process, however, we were not able so far to fully present these new solutions as separate publications – in most part, this is the future work for us, as the patenting procedures are over.

7. What is the novelty (originality) of the review? And What is new in your review that makes a difference in the body of knowledge? 

This review aims to provide, in one place, a state-of-the-art in the field of fluoride-waste management. The novelty of the review is not based on some new methods/solutions proposed. We instead tried to summarize most of the already proposed techniques, pointing out their pro- and contra-, and discussing many aspects, technologies, and possible applications. We also included a description of our own research in this field, patented and published elsewhere.

8. How did you do quality control (QC) and quality assurance (QA) on the obtained secondary data to validate the conclusions?

We do not provide any secondary data to validate the conclusions. What do you mean by QC and QA for non-existing data?

9. What are possible technology-oriented applications of the work for commercialization purposes?

The article gathers in one place several existing and newly-proposed techniques/methods of removing fluoride waste. Thanks to this, it is a good starting point of searching for knowledge on this subject without a need to look for all this information in the literature. Therefore, it allows to find the most interesting method for the reader, and the literature references allow for a broader exploration of knowledge after selecting a specific method.

10. How would this work advance the previous work done in the existing field of study and/or across other fields?

As described above, this work is a single-place review of as many solutions regarding fluoride-waste management as possible, addressing the most advanced and recent proposals in this area. The goal of any review is to help people to compare and search for similar technologies/methods. As such, we do not propose any new method/technique; however, as we compare and point out several external solutions (including our proposals to be published elsewhere and patented), one is able to find the best solution from his/her individual point of view.

11. Based on the data obtained, what are the implications of this work (a) to the field of study, (b) to industry, (c) economy, and/or (c) to the wider public/society in general?

The implications were developed in Section 5.

12. How would the outcomes of work directly contribute to global climate change mitigation and circular economy?

The reuse of waste is a classic example of a circular economy and environment protection.

13. What are the likely research impacts of this work globally, nationally and locally?

We have presented in one place several methods that may be of interest to people involved in the subject. The problem presented in the paper is certainly a global one.

14. What are the economical benefits of this work for industrial purposes?

Economic advantages raise form developing new materials from waste. Their use reduces costs by limiting the use of primary raw materials. Such conclusions are broadened in Section 5.

15. Make a table of comparison of this present work and other similar techniques in terms of operational parameters and operational cost; afterward, please give critical analysis on its technical feasibility and applicability for upscaling this treatment process.

It is such a large topic that it requires a separate discussion. The suggested table will be included in a separate paper that we are currently working on.

  1. Why do you believe your work to be important? What long-term impacts will it have on environmental protection and the wider public or the field following the completion of the project?

This work discusses many methods of neutralizing fluorides. The material we propose allows the management of two different types of waste, creating new materials of a high value. The discussed outcomes of the process of waste treatment may be used, among others, as building materials.

  1. What about the thermal stability of the material obtained from the published studies?

We do not know exactly which material you mean, but there are many publications on thermal properties pointed in the text.

  1. A comprehensive managerial insight should be provided in this paper.

We really do not understand what you mean by “managerial insight” addressed to a scientific publication. Is it something related to a business model for waste management? Or a proposal to establish a new business according to an application of a new technology/solution? Or some law regulations for proper management of ‘dangerous’ compounds? Anyway, all the business-related aspects are, for us, out-of-the-scope of this paper, except maybe for a comparison of costs of competitive methods.

Round 2

Reviewer 2 Report

well done!